# Activity and Stability of Panx1 Channels in Astrocytes and Neuroblastoma Cells Are Enhanced by Cholesterol Depletion

**DOI:** 10.3390/cells11203219

**Published:** 2022-10-14

**Authors:** Antonio Cibelli, Eliana Scemes, David C. Spray

**Affiliations:** 1Department of Neuroscience, Albert Einstein College of Medicine, Bronx, NY 10461, USA; 2Department of Cell Biology and Anatomy, NY Medical College, Valhalla, NY10595, USA

**Keywords:** membrane lipids, brain, cell biology, cholesterol, membrane fluidity, cyclodextrins, Panx1, ATP release, fluorescence recovery after photobleaching, P2X7R

## Abstract

Pannexin1 (Panx1) is expressed in both neurons and glia where it forms ATP-permeable channels that are activated under pathological conditions such as epilepsy, migraine, inflammation, and ischemia. Membrane lipid composition affects proper distribution and function of receptors and ion channels, and defects in cholesterol metabolism are associated with neurological diseases. In order to understand the impact of membrane cholesterol on the distribution and function of Panx1 in neural cells, we used fluorescence recovery after photobleaching (FRAP) to evaluate its mobility and electrophysiology and dye uptake to assess channel function. We observed that cholesterol extraction (using methyl-β-cyclodextrin) and inhibition of its synthesis (lovastatin) decreased the lateral diffusion of Panx1 in the plasma membrane. Panx1 channel activity (dye uptake, ATP release and ionic current) was enhanced in cholesterol-depleted Panx1 transfected cells and in wild-type astrocytes compared to non-depleted or Panx1 null cells. Manipulation of cholesterol levels may, therefore, offer a novel strategy by which Panx1 channel activation might modulate various pathological conditions.

## 1. Introduction

The Pannexins (Panxs) were discovered through their sequence homology to the invertebrate gap junction forming proteins, the innexins [1]. Each of the three Panxs is dynamically regulated during development and differentially expressed in a tissue dependent manner [2,3]. Panx1 forms large conductance channels allowing the passage of molecules up to 1 kDa, mediating ATP and glutamate release, and is capable of dye uptake in many cell types [4,5,6]. In the central nervous system (CNS), Panx1 is expressed in both neurons and glia [7,8]. Dysregulation of Panx1 activity was implicated in several neurodegenerative diseases including Parkinson’s [9] and Alzheimer’s diseases [10], ischemia [11,12], epilepsy [13,14], and in migraine and neuropathic pain [15,16]. Although Panx1 was originally characterized by its activation by membrane depolarization [17], subsequent studies revealed that Panx1 channels can be activated by diverse stimuli including shear stress and hypoosmotic swelling [18,19], receptor mediated pathways (NMDA, P2X7, alpha-adrenergic agonists) and thromboxane [19,20,21,22,23], high extracellular K^+^ [24] and intracellular calcium levels [19], HIV infection [25], and caspase mediated C-tail cleavage [26].

Panx1 channels are integral membrane proteins embedded within the cell plasma membrane consisting of assemblies of complex mixture of phospholipids, glycolipids, and cholesterol [27,28]. Studies in smooth muscle cells showed that Panx1 is enriched in cholesterol- and caveolin-1 (Cav-1)-containing membrane rafts to regulate blood pressure [29]. Brain is the most cholesterol-rich organ, containing about 20% of the whole body’s cholesterol, and essentially all (>99.5%) cholesterol is present in an unesterified form [30]. Cholesterol is an important structural component of cellular membranes and affects a number of properties including lipid raft formation, membrane fluidity, vesicle formation, and synaptogenesis [31,32,33]. Although the half-life of brain cholesterol in adulthood is quite long (between 6 months and 5 years), in the CNS of humans and mice, >95% of cholesterol is synthesized de novo [34,35,36,37]. In contrast to other organs, in which the need for cholesterol is met by cellular uptake of lipoprotein cholesterol, the blood–brain barrier (BBB) prevents the influx of cholesterol-carrying lipoproteins from the circulation. Thus, neurons and mostly astrocytes both synthetize cholesterol de novo using different pathways [38]. Because adult neurons have a lower capacity to make up for a cholesterol deficit by de novo synthesis compared to astrocytes, glial cells play a pivotal role in cholesterol brain homeostasis, synthesizing and transporting cholesterol to neurons via lipoprotein particles [31,39]. Cholesterol levels are tightly controlled in neural cells, and defects in cholesterol metabolism lead to CNS dysfunction and pathologies such as Smith–Lemli–Opitz syndrome [40] and are prominent changes in Huntington’s disease [41] and Alzheimer’s disease [42].

Few studies have examined determinants of Panx1 delivery to the cell membrane and its residence there, although they were shown to be regulated in part by interaction with actin microfilaments [43]. In order to address the distribution of Panx1 and the dynamic organization in specific membrane surface microdomains, their activity and dependence on cholesterol in the cell plasma membrane, we have examined the effects of cholesterol depletion on cell surface mobility of Panx1 and activity of the channels that it forms. For these studies, we monitored the membrane environment of the Panx1 channel by analyzing its lateral diffusion parameters in the plasma membrane of living cells using the fluorescence recovery after photobleaching (FRAP) approach. In light of the significance of cholesterol in the organization and function of membrane proteins, we monitored the effect of the specific cholesterol depletion agents MβCD and lovastatin on the lateral diffusion parameters of Panx1 in N2a neuroblastoma cells and in primary cultures of astrocytes. To evaluate the extent to which cholesterol depletion affected the activation of the Panx1 channel, electrophysiological, dye uptake, and ATP release assays were performed. Our findings collectively reveal that cholesterol depletion induces a lower Panx1 mobility in the membrane and directly enhances its activity.

## 2. Materials and Methods

### 2.1. Animals and Ethics Statement

Post-natal 1–3 day old Panx1^tm1a (KOMP) Wtsi^ (global Panx1-null) mice in the C57BL/6 background [44] and age matched controls (Panx1^f/f^) were used to obtain astrocyte primary cultures. All procedures described below were reviewed and approved by the Einstein Institutional Animal Care and Use Committee (IACUC). The procedures follow ARRIVE guidelines and are consistent with the Guide for the Care and Use of Laboratory Animals, 8th Edition. Mouse colonies were maintained in AALAC-accredited Laboratory Animal Resource Facility with a 12-h light/dark cycle (lights on at 7:00 am, light off at 7:00 pm) and with access to food and water ad libitum. 

### 2.2. Cell Culture 

These studies used the Neuro2A neuroblastoma (ATCC CCL-131, here termed N2a) cell line and primary cultures of mouse cortical astrocytes. N2a cells were grown in high glucose DMEM with 10% fetal bovine serum (FBS) and 1% antibiotics (100 U/mL penicillin and 10 mg/mL streptomycin). N2a cells lacking Panx1 were obtained through deletion of endogenous mPanx1 through CRISPR technology (described below). Primary cultures of Panx1 null and Panx1^f/f^ astrocytes [44] were obtained from perinatal C57Bl6 transgenic mice lacking Panx1, using our standard methods [45]. In brief, after a painless death, the brains of 1 to 3-days old mice were removed and stripped of meninges. Cortices were dissected and maintained in ice-cold phosphate buffered saline (ThermoFisher, Waltham, MA, USA), then the tissues were chopped into small pieces with a sterile scalpel and chemically dissociated using 0.05% Trypsin EDTA for 20 min at 37 °C in a 95% air, 5% CO_2_ incubator. Thereafter, cells were centrifuged, and the supernatant was re-suspended in DMEM-Glutamax medium (ThermoFisher, Waltham, MA, USA) with 10% FBS, 1% antibiotics, and maintained at 37 °C in a 95% air, 5% CO_2_ incubator. After 15 days in culture, cells other than astrocytes were removed by shaking for 4 h at 180 rpm at 37 °C. Astrocytes were then cultured in DMEM + 10% FBS + antibiotics and used at one week after plating.

### 2.3. Generation of Panx1-Null N2a Clones through CRISPR-Cas-9 Editing

For CRISPR technology, we used the services of the Gene Modification Facility at Albert Einstein College of Medicine. Briefly, guided (g) RNA targeting exon1 and gRNA targeting exon2 of mouse *Panx1* gene were designed by an online tool (https://benchling.com/, accessed on 1 March 2017) and used to generate the corresponding gRNA/cas9 all-in-one plasmids by SLiCE [46], termed Px330-puro-*Panx1* Ex1 52/84 and Px330-puro-*Panx1* Ex2 61/64. N2a cells were co-transfected with gRNA/cas9 all-in-one plasmid Px330-puro-*Panx1* Ex1 52/84 and Px330-puro-*Panx1a* Ex2 61/64 using Lipofectamine 3000 reagent (Invitrogen, Waltham, MA, USA) according to manufacturer’s instructions. Around 48 h after transfection, puromycin (5.0 µg/µL) was added for a duration of 48 h. The resulting puromycin resistant cell population was collected and assayed for CRISPR cleavage efficiency. Genomic DNA was extracted from the resulting puromycin resistant cell population. PCR of the adjacent regions of the gRNA/Crispr targeting sites was performed using primers with sequences given below. PCR products were then cloned into a plasmid vector, PBluscript II KS+, by SLiCE. Multiple colonies were picked and subjected to Sanger sequencing with primer M13 Rev (located on the vector backbone). All the primers contain 30 nt of overlapping sequences with plasmid vector pBluscript II KS+ at 5′ ends for SLiCE cloning and 20 nt binding sequence specific to *Panx1* gene at 3′ ends.

### 2.4. PCR Primers 

PBE-*Panx1* Ex1 GTF CCGGGCTGCAGGAATTCGATggccacggagtatgtgttct

PBE-*Panx1* Ex1 GTR ACGGTATCGATAAGCTTGATgcaggagacaaaggaactgg 

PBE-*Panx1* Ex2 GTF CCGGGCTGCAGGAATTCGATtgtgatgttgcagctcagtg 

PBE-*Panx1* Ex2 GTR ACGGTATCGATAAGCTTGATcctctggcatatcgtggact

### 2.5. Generation of Stable hPanx1-Expressing N2a Cells

Recombinant lentivirus particles pLV[Exp]-mcherry:T2A-Puro-EFS (VectorBuilder, Chicago, IL, USA) were used to express *hPanx1* (NM_015368.3) in Panx1-deleted N2a cells. Cells were infected using three multiplicities of infections (MOI: 2.5, 5.0, and 10.0) and cells maintained for 24 h at 37 °C in a humidified 5% CO_2_ incubator. After removal of medium containing the virus, cells were maintained in regular DMEM for 24 h and then maintained in DMEM containing puromycin for selection of clones expressing the mCherry marker, visualized using an epifluorescence microscope.

### 2.6. Transfection with Other Plasmids

Panx1 deleted N2a cells were transiently transfected with [*mPanx1*-eGFP (pLenti6/V5-EF1a-sfGFP-N1 *MPanx1*CDS)] using TransIT-LT1 (Mìrus, LLC, Madison, WI, USA) transfection reagent according to manufacturer’s instructions in Opti-MEM media, which was then replaced with the standard growth media for astrocytes and N2a cells (DMEM with 10% FBS and 1% antibiotics) 12–16 h after transfection. 

### 2.7. Immunocytochemistry and Image Acquisition

Parental, Panx1-deleted and stable hPanx1 transfected N2a cells plated on glass bottom dishes were immunostained for Panx1. Cells fixed in 4% p-formaldehyde (PFA) were permeabilized with 0.2% Triton X-100 for 15 min, then incubated for 30 min in a blocking solution (2% BSA in PBS) and then overnight with rabbit anti-Panx1 antibody (1:200; Alomone, Jerusalem, Israel; cat#ACC-234) in blocking solution. After washout of the primary antibody, cells were exposed to donkey anti-rabbit secondary antibody (1:1000; Alexa Fluor 488, Invitrogen, Waltham, MA, USA; cat#A-21206) for 1 h at room temperature along with DAPI to stain nuclei, prior to mounting (ProLong Gold antifade, Invitrogen cat#P36930). Images were captured with an automated inverted Leica DMi8 SP8 microscope (Wetzlar, Germany) using a 63× NA = 1.4 Oil PLAPO, WD = 0.14 mm liquid immersion objective with a refractive index of 1.5. Leica LASX software (version number 3.3.0, Wetzlar, Germany) was used for image acquisition and analysis (Leica Microsystems CMS GmbH). Confocal images were collected using 488 nm laser beam and a pinhole diameter of 1 Airy unit.

### 2.8. Treatments with Cyclodextrin and Lovastatin

N2a cells and astrocytes were depleted of cholesterol by two alternative methods: MβCD treatment to extract cholesterol from the membrane and lovastatin to inhibit its synthesis. For MβCD (Sigma, Saint Louis, MO, USA; cat#C4555) treatment, cells were washed with PBS and incubated with 5 or 10 mM MβCD in serum free DMEM for 60 min at 37 °C. Cholesterol enrichment or repletion of cholesterol-depleted cells were obtained by incubation of the cells for 2 h at 37 °C using MβCD-cholesterol inclusion complexes (formed by mixing cholesterol suspension with cyclodextrin solution, as described [47]). To decrease cholesterol synthesis, we applied lovastatin, an inhibitor of 3-hydroxy-3-methylglutaryl-coenzyme A (HMG-CoA) reductase, the enzyme that catalyzes the conversion of HMH-CoA to the cholesterol precursor mevalonate. For studies with lovastatin, the cells were incubated 12 h with 1, 5, or 20 µM lovastatin (Sigma) in serum free DMEM at 37 °C.

### 2.9. Cell Viability Assay

Cell viability was determined using 3-(4,5-dimethylthiazol-2-yl)-2, 5-diphenyltetrazolium bromide tetrazolium (MTT) reduction assay on N2a mPanx1-GFP transfected cells. The cells were seeded in 96-well plates at 3000–5000 per well. After overnight incubation, cells were treated with 5 mM and 10 mM MβCD or with the complex MβCD–cholesterol for 1 h and 2 h, respectively, or pre-treated with MβCD for 1 h and then treated with cholesterol for 2 h. Control groups were incubated with non-supplemented media at the same time and processed similarly. Next, treatment solutions were removed and 100 µL MTT (0.5 mg/mL of MTT in serum-free medium) was added to each well for 2 h at 37 °C. After incubation, 100 µL of DMSO was added into each well for 15 min to dissolve the formazan crystals. The absorbance was measured at 570 nm with a FLUOstar Omega Microplate Reader (BMG Labtech, Ortenberg, Germany). All experiments were repeated at least three times in triplicate. Cell viability was expressed as the percentage of control.

### 2.10. Analysis of Cholesterol Content

N2a cells grown overnight (~75–90% confluence) were transfected with mPanx1-eGFP in 6-well plates. Fourteen hours post-transfection, media was replaced with DMEM +10% FBS. Twenty hours post-transfection, FBS was removed, and cells were untreated or exposed to 5 mM or 10 mM MβCD or with the MβCD-cholesterol complex for 1 h and 2 h, respectively, or pre-treated with MβCD for 1 h and then treated with cholesterol for 2 h. The cholesterol content in cell membranes was determined using the Amplex Red cholesterol assay kit in accordance with the manufacturer’s protocol (Thermo, Waltham, MA, USA). Briefly, cell homogenates were incubated with the reaction mix (0.2 U/mL cholesterol esterase, 2 U/mL cholesterol oxidase, 2 U/mL HRP, 300 mM Amplex Red) for 30 min in the absence of light at 37 °C. Lysis buffer was used as negative control, and for the positive control, serial dilutions of cholesterol (0–0.8 µg/mL) were used. Fluorescence was measured with FLUOstar Omega Microplate Reader (BMG Labtech, Ortenberg, Germany) using an excitation wavelength of 560 nm and emission wavelength of 590 nm.

### 2.11. Panx1 Extraction by MβCD 

N2a mPanx1-eGFP expressing cells were seeded in 6-well plates at 3000 to 5000 cells per well. After overnight incubation, FBS was removed, and cells were untreated or exposed to 5 mM and 10 mM MβCD or with the complex MβCD-cholesterol for 1 h and 2 h, respectively, or pre-treated with MβCD for 1 h and then treated with cholesterol for 2 h. After treatments, media were collected and centrifuged at 15,000 rpm for 1 min, and 100 μL was added to black-walled 96-well plates in triplicate. eGFP fluorescence was measured at an excitation wavelength of 485 nm and emission wavelength of 528 nm using the FLUOstar Omega Microplate Reader; extracted protein in each treatment group was calculated as fluorescence relative to the corresponding control group.

### 2.12. ATP Release

Confluent cultures of Panx1^f/f^ and Panx1-null astrocytes plated in 35 mm dishes in the absence of serum were treated with 5 mM MβCD for 1 h or 10 μM lovastatin for 16 h at 37 °C in a 5% CO_2_ incubator. For measurements of ATP released following MβCD incubation, aliquots of the medium were collected 60 min after the treatment. For measurements of ATP released following lovastatin treatment, samples were collected 16 h after the treatment. For each time point, 100 μL samples were added to black-walled 96-well plates in triplicate and ATP content was measured using the luciferin/luciferase assay (Invitrogen) and the FLUOstar Omega Microplate Reader, in accordance with the manufacturers protocol. The amount of ATP in each sample was normalized to the total protein concentration, using the BCA assay (Pierce).

### 2.13. Dye Uptake

To quantify the opening of Panx1 channels in cholesterol depleted cells, N2a (Panx1-null and stably hPanx1-transfected) and astrocytes (Panx1^f/f^ and Panx1-null) plated on glass bottomed dishes (~75–90% confluence) were incubated with MβCD or lovastatin as described above. After PBS washing, treated cells and untreated controls were bathed for 5 min in divalent-free (zero Ca^2+^/Mg^2+^) phosphate buffered solution (dPBS, pH 7.4) containing the cell-impermeant dye ethidium bromide (EtBr, 10 μM; Sigma [48]) in the absence and presence of 100 μM BzATP (Sigma). After removal of the dye containing solution, cells were fixed in 4% PFA for 5 min at room temperature. Cells were examined with a Nikon Eclipse TE-3000E microscope (Nikon, Tokyo, Japan) coupled to a SPOT RT CCD camera. Images were acquired with a 20× objective using Spot32 software and analyzed with ImageJ software (version number 1.8.0, National Institutes of Health, USA). The influx of EthBr in cholesterol depleted N2a cells (Panx1-null and stably hPanx1-transfected) and astrocytes (Panx1^f/f^ and Panx1-null) was determined in BzATP treated cells, with the uptake calculated as the dye intensity value relative to that measured under control conditions (untreated). At least three random fields were selected in each coverslip.

### 2.14. Electrophysiology

To evaluate the electrophysiological activity of Panx1 channels in cholesterol depleted cells, N2a (Panx1-null and stably hPanx1-transfected) cells plated on glass coverslips were incubated with 5 mM MβCD or 10 μM Lovastatin 1 h or 12 h prior to recordings. Whole cell patch clamp recordings were performed as previously described [23] on cells that were eGFP positive. Briefly, cells were bathed in phosphate buffered saline (PBS) containing 10 mM HEPES and 1 g/L glucose (composition in mM: NaCl 147, HEPES 10, CaCl_2_ 2, MgCl_2_ 1 and KCl 2.7, 8 mM Na_2_HPO_4_, and 2 mM KH_2_PO_4_ pH 7.4). The pipette solution contained (mM): CsCl 130, EGTA 10, HEPES 10, CaCl_2_ 0.5, pH 7.4. Activation of Panx1 channels by voltage was achieved by applying 1 s −10 mV pulse, followed by 12 s voltage ramps from a holding potential of −60 mV to +80 mV. Electrophysiological recordings were accomplished using an Axopatch 1-C amplifier, and pClamp10 software (San Jose, CA, USA) was used for data acquisition and analysis. Differences in peak conductance induced by MβCD or lovastatin were normalized to those recorded in untreated cells and expressed as fold changes.

### 2.15. Fluorescence Recovery after Photobleaching (FRAP)

For FRAP experiments, Panx1-transfected N2A cells or primary cultures of transfected astrocytes were plated into 35 mm imaging chambers (ibidi, Gräfelfing, Germany; cat no. 80136) at ~70–80% confluency and treated with MβCD or lovastatin. Next, treatment solutions were removed and replaced with imaging medium containing DMEM without phenol red. Then, 2D time-lapse imaging was conducted as described previously [49,50] in a chamber at room temperature or at 37 °C using a Zeiss LSM 510 Live with Duo module and imaged with a 63×, 1.4 numerical aperture oil immersion objective. Cells were imaged at 500 ms intervals with 489 nm laser illumination for both image acquisition and bleaching. For bleaching, the laser power was set to 100% and directed to a rectangular region of 10 × 5 µm with a scan speed of 5 frames/s and 3 bleach iterations.

### 2.16. FRAP Data Analysis

Fluorescence recovery curves were calculated from regions of interest (ROIs) corresponding to the bleach regions (fp): the fluorescence pool available for recovery (Fp) and the background fluorescence (bF, a location with no GFP expression) and corrected for loss of overall signal and normalized to 100% of fluorescence intensity before (prebleach), and to 0% of fluorescence intensity at the first time point post-bleach, as described previously [49]. 

FRAP% = (fpFo/Fp) × (bF/bFo) × 100%.

The normalized data points at 15, 30, or 60 s after the bleach were used in comparison of percent recovery at 15, 30, or 60 s.

### 2.17. Statistical Analysis

All experiments were carried out at least three times and the results are presented as a mean ± standard error of the mean (SEM). Statistical analysis was performed using RStudio and GraphPad7 software (San Diego, CA, USA. First, a Shapiro–Wilk normality test was performed on the data, to evaluate the data distribution, then, statistical significance was evaluated using the unpaired Mann–Whitney test for non-parametric data, while the unpaired Student’s *t* test or ANOVA with post hoc analysis (Newman–Keuls and Dunnett’s test) were performed for normally distributed data. Significance was considered *p* < 0.05 and designated on figures as follows: * *p* < 0.05, ** *p* < 0.01, *** *p* < 0.001, and **** *p* < 0.0001.

## 3. Results 

### 3.1. Endogenous Panx1 Is Markedly Reduced in CRISPR-Deleted N2a Cells 

The mouse N2a neuroblastoma cell line expressed endogenous Panx1, as evidenced by immunostaining (Figure 1A). For studies described below using N2a exogenously expressing hPanx1, we first generated an N2a-Panx1-null cell line using Crispr/Cas-9 technology. Using this approach, we obtained successful knockdown of Panx1. Immunocytochemistry revealed complete knockdown of Panx1 in the vast majority of the N2a cells although a few still displayed low levels of Panx1 immunoreactivity (Figure 1B). Panx1-deleted cells infected with hPanx1 displayed substantial Panx1 immunofluorescence (Figure 1C).

### 3.2. Effects of Temperature on Panx1 Mobility

Temperatures in the physiological range generally have profound effects on physical properties of membrane lipids. In order to determine the impact of temperature on membrane mobility of Panx1, we compared FRAP experiments performed at 37 °C and 25 °C. As shown in the series of images in Figure 2A and plotted in Figure 2C,D, photobleach recovery at 37 °C after one minute was very rapid (69.7 ± 3.7%) compared with recovery curves at the lower temperature (49.6 ± 1.2%) (Figure 2B). This enhanced mobility of Panx1 at 37 °C suggests that higher temperature modifies membrane fluidity, promoting lateral movements in the membrane.

### 3.3. Effect of Cholesterol Treatments on Membrane Cholesterol Level and Cell Viability of N2a Cells

To determine the extent to which methyl beta cyclodextrin (MβCD) and lovastatin reduced cholesterol in N2a cells, we measured total cholesterol levels after treatments. As shown in Figure 3A, cholesterol was strongly depleted by one hour treatment with 5 and 10 mM MβCD (reduced by 38.7 ± 3.3% and 62.6 ± 2.7%, respectively), was enriched by 2 h treatment with cholesterol alone (by 28.2 ± 1.7%), and was restored to 119.5 ± 3.5% by 2 h cholesterol treatment after 1 h MβCD extraction (5 mM). Exposure to lovastatin showed a cholesterol depletion of 16.2 ± 2.6%.

To determine if extraction of cholesterol by MβCD and lovastatin affected levels of Panx1, we measured the GFP fluorescence in the medium of N2a mPanx1-GFP transfected cell after treatments. As shown in Figure 3B, no differences were detected in the GFP fluorescence in the medium after cholesterol depletion or after cholesterol replenishment compared to the untreated control. In addition, no change was detected in total GFP fluorescence within the cells (Appendix A). This indicates that extraction of cholesterol with MβCD or lovastatin does not lead to appreciable extraction of mPanx1 from the cell membranes (Figure 3B).

To evaluate the potential toxicity of MβCD on N2a mPanx1-GFP transfected cells, we measured cell viability after cholesterol modifying treatments. As shown in Figure 3C, 5 mM MβCD did not significantly affect cell viability (92.6 ± 3.5% cells viable), whereas at a higher concentration (10 mM), MβCD significantly increased cell death (26.96 ± 1.3% cell viability; *p* < 0.01 ANOVA followed by Dunnett’s multiple comparisons test). Thus, the lower MβCD concentration was used in most subsequent cholesterol depletion experiments. Exposure of N2a cells to the MβCD-cholesterol complex and lovastatin revealed no cytotoxicity (93.3 ± 5.1% and 95.2 ± 3.1% viable, respectively); however, pre-treatment with 5 mM MβCD (1 h) and then incubation with MβCD-cholesterol complexes for cholesterol repletion (2 h) induced a partial reduction in cell viability (45.86 ± 2.8%; *p* < 0.05 ANOVA followed by Dunnett’s multiple comparisons test).

### 3.4. Effects of MβCD on the Lateral Diffusion of mPanx-1-GFP in Cell Membrane

We applied photobleaching protocols to measure Panx1 diffusion in membranes of N2a cells expressing GFP-tagged Panx1 under normal conditions and after cholesterol depletion and repletion. Under control conditions, photobleach was followed by a fairly rapid fluorescence recovery, reaching 55 ± 2.4% of the original pre-bleach condition within 1 min (Figure 4A, quantified in Figure 4D,E). By contrast, FRAP after cholesterol depletion revealed substantially lower Panx1 mobility in the membrane (Figure 3B), with only 28 ± 2.4% recovery at one min postbleach, as quantified in Figure 4D,E. The repletion in cholesterol in MβCD depleted cells led to FRAP recovery with kinetics similar to those recorded in untreated controls (Figure 4C–E). The increasing MβCD concentration to 10 mM further slowed recovery kinetics (17.3 ± 2.1% at 1 min postbleach). A delayed FRAP recovery was also observed following treatment with the MβCD analog 2-Hydroxypropyl-β-cyclodextrin, where 50 mM led to virtually complete immobility (Appendix A).

### 3.5. mPanx1-GFP Mobility at Plasma Membrane in Panx1-Null Astrocytes

To determine whether the impact of cholesterol depletion on Panx1 mobility seen in N2a cells was similar in the primary cultured astrocytes, we transfected astrocytes isolated from Panx1-null mice with GFP-tagged mPanx1 and measured Panx1 mobility following cholesterol depletion or replenishment. As in the case of the N2a, mobility was slower following cholesterol depletion (with only 29.9 ± 1.7% recovery at one min postbleach compared to 46.1 ± 1.4% in untreated cells) and the FRAP recovery rate was restored (43.7 ± 1.2% recovery at one min postbleach) by addition of cholesterol. The results of these studies are summarized in Figure 5A,B and reveal a strong similarity to results obtained with N2a cells.

### 3.6. Panx1 Mobility after Inhibition of Cholesterol Synthesis

As an alternative method to lowering cholesterol levels without using a compound that extracts cholesterol from the membranes, we tested the effects of lovastatin, an inhibitor of cholesterol synthesis. As illustrated in Figure 6A,B, FRAP recovery of GFAP-tagged mPanx1 N2a cells treated for 12 h with 5 µM lovastatin was attenuated (with only 18.6 ± 1.9% recovery at one min postbleach compared to 46.7 ± 1.3% recovery in untreated cells). FRAP experiments following 12 h treatment with 1, 5, and 20 µM lovastatin revealed that higher concentrations progressively reduced extent and rate of recovery (Figure 6D). These studies show that inhibition of cholesterol synthesis results in substantially reduced Panx1 mobility, confirming the MβCD results.

### 3.7. Panx1 Channels Mediate Dye Uptake in Cholesterol-Depleted Cell Plasma Membrane

Panx1 channels mediate diffusive release and uptake of molecules with MW up to about 1 kDa. To determine the impact of cholesterol depletion on this property of Panx1 channels, we measured the uptake of EthBr in mPanx1 transfected N2a cells under control conditions and following cholesterol depletion (1 h treatment with 5 mM MβCD and 12 h with 5 µM lovastatin). As shown in the upper row of panels in Figure 7A and quantified in Figure 7B, under control conditions, EthBr uptake was increased by cholesterol depletion as above, with a 1.3-fold increase after MβCD and 1.6-fold increase by lovastatin. To determine if dye uptake was also altered under conditions when Panx1 channels were activated by another stimulus, we compared the impact of cholesterol depletion on uptake of dye after treatment with 100 µM BzATP. In this case (Figure 7A, lower row of panel), cholesterol depletion did not lead to further enhancement of dye uptake, implying that BzATP treatment maximally activated the uptake. To confirm that uptake was through Panx1 channels, we performed a parallel series of experiments in N2a cells lacking Panx1. As shown in Figure 7C,D, none of the treatments in Panx1 null cells increased EthBr uptake above baseline levels.

### 3.8. Dye Uptake in Astrocytes 

To determine whether cholesterol depletion also increased EthBr uptake in primary astrocytes, we compared uptake in primary cultures from Panx1^f/f^ and Panx1 null mice. As shown in Figure 8A and quantified in Figure 8C, in Panx1^f/f^ astrocytes, EthBr uptake was increased 1.3-fold after MβCD (1 h, 5 mM) and 2-fold after BzATP treatment (5 min, 100 μM); as in Panx1 transfected N2a cells, combined treatment with BzATP and MβCD did not further increase dye uptake, implying saturation. Involvement of Panx1 in the dye uptake is shown using Panx1 null astrocytes, in which case, none of the treatments increased EthBr uptake above baseline levels (Figure 8B,D).

### 3.9. Cholesterol Sensitivity of N2a Panx1 Currents

Panx1 channels are activated by membrane depolarization, with currents outwardly rectifying at potentials above 0 mV. To determine if Panx1 currents were affected by cholesterol depletion, we compared hPanx1 transfected N2a cells untreated or treated with 5 mM MβCD or 5 µM lovastatin for 1 and 12 h, respectively. As shown in Figure 9A,D for MβCD and in Figure 9B,E for lovastatin, outwardly rectifying currents were greatly enhanced by cholesterol depletion, with a 2.2-fold increase after MβCD and 2.6-fold increase for lovastatin. In Panx1 null N2a cells, in contrast, MβCD treatments did not increase conductance above baseline levels (Figure 9C,F).

### 3.10. Panx1 Channels Mediate ATP Release in Cholesterol-Depleted Cell Plasma Membrane of Cultured Astrocytes

Results described above (Figure 7, Figure 8 and Figure 9) indicate that cholesterol depletion increases dye influx and ionic currents through Panx1 channels. A major function attributed to Panx1 is the release of ATP, with both beneficial and detrimental consequences depending on context. To test for action of cholesterol depletion on ATP release, we compared release of ATP from WT and Panx1-null astrocytes following cholesterol depletion with MβCD and with lovastatin treatment. As illustrated in Figure 10A, ATP release was significantly higher at 1 h with MβCD (1.8-fold increase compared to control) and, as shown in Figure 10C, it was significantly higher at 12 h after lovastatin treatment (1.5-fold increase compared to control). To confirm that uptake was mediated through Panx1 channels, we performed a parallel series of experiments in Panx1-null astrocytes. As shown in Figure 10B,D, none of the treatments in Panx1-null astrocytes increased ATP release above baseline levels. 

## 4. Discussion

We found that Panx1 tagged with fluorescent protein was highly mobile in the plasma membrane of transfected N2a cells and of astrocytes. Recovery kinetics and mobile fraction were comparable to those seen for mobility of other proteins of similar size in a non-junctional membrane (e.g., occludin and Cx30 connexons [51]). Previous FRAP studies performed at room temperature on GFP-tagged mouse Panx1 expressed in a breast cancer cell line (BICR-MIRk) [43] and zebrafish Panx1a expressed in N2a cells [52] revealed diffusion kinetics comparable to those reported here, with recovery of 45–60% at one min after bleaching.

The impact of cholesterol concentration on lateral mobility of other membrane-embedded proteins was reported previously in studies motivated in part by the question of whether partitioning the proteins into cholesterol-rich lipid rafts affects their properties. In a study comparing several raft-localized and non-raft membrane proteins, lateral diffusion coefficients were found to vary over a range of approximately 10-fold, regardless of whether the proteins were raft-associated or not, from which the authors concluded that other constraints such as binding and crowding are more influential determinants of membrane mobility [53]. Treatment with MβCD slowed lateral diffusion of each of the proteins in that study but did not alter mobile fraction, indicating that lipid environment or cytoskeletal attachment affected mobility but that the treatments did not alter distribution of the membrane proteins between mobile and immobile membrane pools.

Early biochemical studies on Panx1 exogenously expressed in cell lines did not detect immunoprecipitation with caveolin1 or altered distribution when cells were exposed to MβCD for up to 25 h, indicating that it was not bound to cholesterol-rich raft domains [54]. However, more recent structural studies have identified leucine residues in Panx1 that directly interact with cholesterol [55]. Subsequent studies on neuroblastoma cells have further revealed that treatment with MβCD interferes with ATP-induced Panx1 internalization, implying a role of cholesterol-rich domains in the process [56]. Moreover, in vascular tissues, binding to caveolin1 within specialized lipid raft domains is hypothesized to position Panx1 for interactions with sodium channels in endothelial cells [57] and with alpha adrenergic receptors in smooth muscle, enhancing local release of purinergic signals [29].

Functional consequences of cholesterol depletion in our studies of both N2a cells and astrocytes included enhanced currents in electrophysiological recordings and more extensive uptake of ethidium bromide, changes not seen in cells lacking Panx1. Quantification of the amount of ATP released from cholesterol depleted wild-type astrocyte cultures provided further evidence that depletion enhances function, and that ATP release is through Panx1 channels. Altered function of other channel proteins though manipulation of cholesterol was reported. For example, water flux through both M1 and M23 isoforms of Aquaporin4 (AQP4) in lipid bilayers was lower in higher cholesterol concentrations, simulating sequestration into lipid rafts that could occur due to increased cholesterol under ischemic conditions [58]. For another well-studied membrane protein, the dopamine transporter, MβCD, increased lateral mobility and both ligand affinity and transport function, from which it was concluded that association with lipid-raft domains served to regulate both distribution and function of this transporter [59]. Furthermore, the effect of MβCD on TRPV4 activity was recently reported in trabecular meshwork cells [60]. Although no data were reported on TRPV4 lateral mobility after cholesterol depletion, they showed that MβCD increased TRPV4 agonist activation, swelling-induced calcium signaling after exposure to hypotonic stimuli and actin expression. Panx1 cell surface localization and mobility are, in part, regulated by actin microfilaments [43], thus, our finding that cholesterol depletion reduces the Panx1 mobility suggests that the increase in actin expression may stabilize and anchor the Panx1 in the cell plasma membrane.

In our studies, cholesterol depletion enhanced dye uptake and membrane currents in response to agonist stimulation. A similar enhancement of EthBr uptake and membrane currents by MβCD was reported previously, although lack of blockade by carbenoxolone in that study indicated that Panx1 was not required [61]. The direct inhibition of P2X7 receptor channels by cholesterol is supported by studies on purified panda P2X7 receptors, in which binding to the transmembrane domain decreased activity [62].

As stated previously, cholesterol plays an essential role in brain development and CNS functions, with astrocytes providing most cholesterol synthesis. Cholesterol metabolic defects lead to numerous pathological processes such as memory and cognitive disorders and neurodegenerative diseases such as Alzheimer’s disease and Huntington’s disease. We here show that cholesterol depletion reduces the membrane mobility of Panx1 and increases its activation. Thus, ATP released from astrocytes through Panx1 may have deleterious consequences for brain activity, altering synaptic activity and increasing susceptibility to death in neurons but also dysregulating the production and the release of cytokines from microglia. In this regard, cholesterol depletion in CNS was extensively reported in human affected by Huntington’s disease but also in murine models of Huntington’s disease [63] where increased P2X7 receptor expression and ATP were reported [64]. Furthermore, cholesterol levels can be acutely altered by viral infection through activation of the interferon stimulated gene cholesterol 25-hyroxylase [65]. Panx1 activation, which is enhanced by cholesterol depletion, intensifies the inflammatory response to SARS-CoV-2 infection in human airway epithelial cells [48].

In addition to its effects on channels embedded in the membrane, cholesterol may act through lipid-mediated signaling in response to CNS injury and neurodegenerative diseases. In this regard, Guttenplan and colleagues showed that reactive astrocytes upregulate saturated lipids and secretion of lipoproteins, producing neurotoxic effects that drive the death of oligodendrocytes and neurons [66]. Thus, the data reported in this paper and those of others indicate that, in a pathological condition, glia cells can release many signaling molecules, including protein [67], ATP, or lipids [68] that may cause toxic influence on the CNS.

## Figures and Tables

**Figure 1 cells-11-03219-f001:**
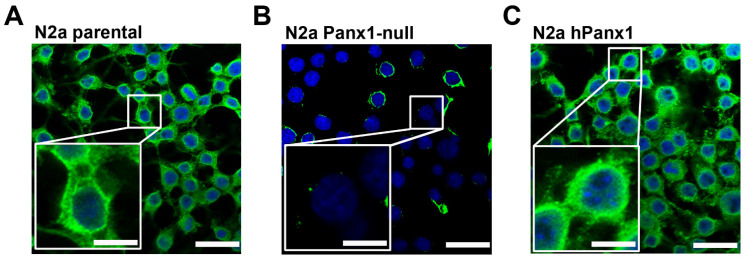
Expression of endogenous Panx1 is largely absent in CRISPR-deleted N2a cells. Representative confocal images obtained from (**A**) parental, (**B**) Panx1-deleted, and (**C**) Panx1-deleted N2a cells expressing human (h) Panx1 construct. Boxed white areas show the Panx1 staining at higher magnification. Scale bars: 50 μm lower magnification, 20 μm in boxed areas.

**Figure 2 cells-11-03219-f002:**
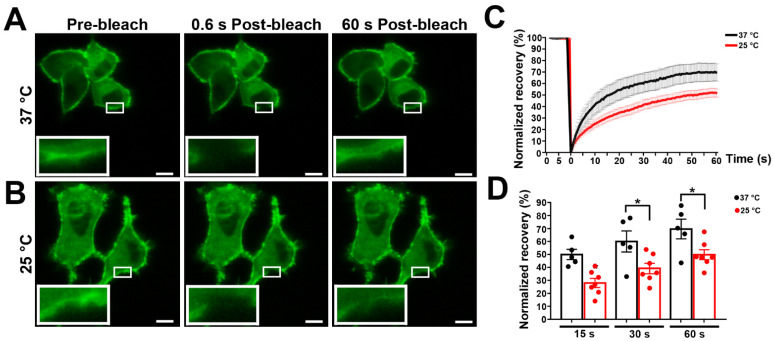
Effects of temperature on the lateral diffusion of mPanx1-GFP protein. N2a cells transiently expressing mPanx-1-GFP were subjected to FRAP at 37 °C (**A**) and 25 °C (**B**). Images were obtained at indicated time points after photobleaching. Magnification scale bars: 10 µm; insets show FRAP regions. (**C**) Cumulative FRAP curves showing more rapid recovery at 37 °C than 25 °C. (**C**,**D**) Histograms of normalized FRAP recovery at 15, 30, and 60 s after photobleach. Each curve represents the average of 7–10 cells. *p* values were obtained using unpaired Student’s *t* test. Data collected over three independent repeats. Bars represent mean ± SEM. * *p* < 0.05.

**Figure 3 cells-11-03219-f003:**
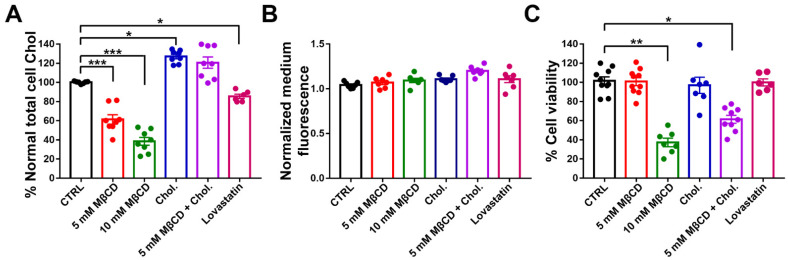
Effects of treatment with MβCD and lovastatin on membrane cholesterol level, Panx1 extraction, and cell viability. (**A**) mPanx-1-GFP transfected N2a cells were incubated with cyclodextrin (5 mM MβCD and 10 mM MβCD for 60 min at 37 °C), lovastatin (5 μM, 12 h) in serum-free DMEM or with cholesterol (2 h, 37 °C). Washed membrane fractions were assayed for cholesterol with results normalized with respect to the protein content of the membrane fraction. Bars show mean ± SEM of replicates of three experiments. (**B**) Effects of MβCD and lovastatin in medium of Panx1 cells. (**C**) Effects of MβCD, lovastatin, and cholesterol on cell viability of mPanx1-GFP transfected N2a cells measured by MTT assay after treatments. *p* values obtained from ANOVA test followed by Dunnett’s test. Results are representative of three independent experiments ± SEM.* *p* < 0.05, ** *p* < 0.01, *** *p* < 0.001.

**Figure 4 cells-11-03219-f004:**
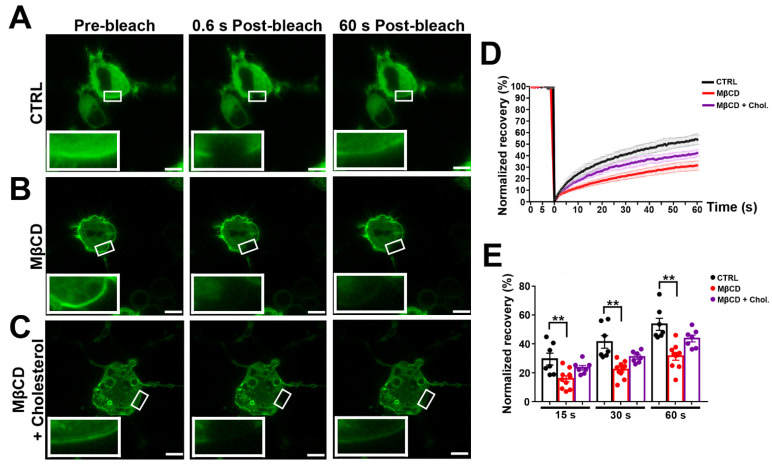
Cholesterol depletion reduces lateral mobility of Panx1 in the membrane. N2a cells transiently expressing mPanx-1-GFP were subjected to treatment with 5 mM methyl-β-cyclodextrin (MβCD), 5 mM MβCD followed by addition of cholesterol (MβCD + Chol) or left untreated (CTRL). (**A**–**C**) Fluorescence images of transfected cells (control and treated with MβCD alone or with cholesterol) before (pre-bleach) and at 0.6 and 60 s postbleach; bleached regions are shown at higher magnification in insets. Bars: 10 μm; inset magnification 3×. (**D**) FRAP recovery curves for control cells and with MβCD alone or with cholesterol. (**E**) Histograms of normalized FRAP recovery at 15, 30, and 60 s after photobleach. *p* values obtained from ANOVA test followed by Dunnett’s test. *n* = 7–10 per plasma membrane domain, data collected over four independent repeats. ** *p* < 0.01.

**Figure 5 cells-11-03219-f005:**
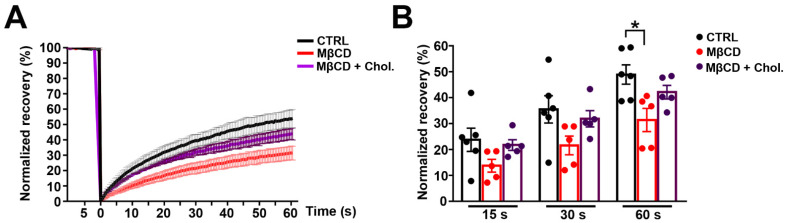
Cholesterol depletion reduces lateral diffusion of mPanx-1-GFP in Panx1-null astrocytes. (**A**) Panx1-null astrocytes transiently expressing mPanx-1-GFP were subjected to treatments with 5 mM methyl-β-cyclodextrin (MβCD), 5 mM MβCD followed by addition of cholesterol (MβCD + Chol) or left untreated (CTRL). Typical FRAP curves of the lateral diffusion of mPanx1-GFP and its modulation by the various treatments are depicted in A. mPanx1-GFP lateral diffusion rates were reduced by MβCD (**B**). Normal diffusion was restored when cholesterol was added. (* *p* < 0.05). Bars: 10 μm. *p* values obtained from ANOVA followed by Dunnett’s test. *n* = 7–10 per plasma membrane domains; data collected over four independent repeats.

**Figure 6 cells-11-03219-f006:**
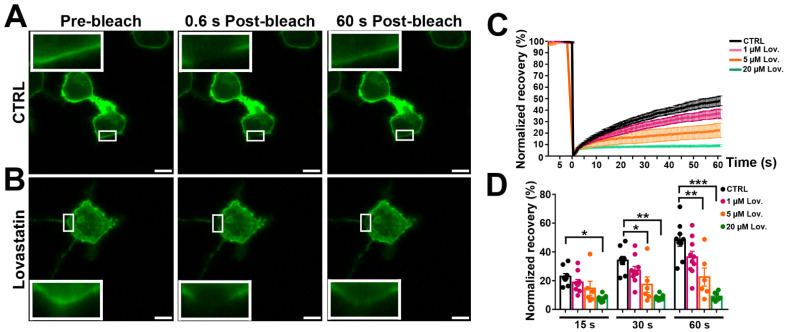
Cholesterol depletion by lovastatin impairs lateral mobility of mPanx1-GFP. (**A**) Fluorescence micrographs of N2a cells transiently transfected with mPanx1-GFP showing FRAP results before and after FRAP in normal medium and (**B**) after 12 h treatment with 5 µM lovastatin. Bars: 10 μm; insets in each panel are higher magnification of FRAP area (3×). (**C**) FRAP recovery curves obtained after treatment with 1, 5, and 20 µM lovastatin. (**D**) Normalized FRAP recovery at 15, 30, and 60 s after photobleach. *p* values obtained from ANOVA followed by Dunnett’s test. Each curve represents the average from 7 to 10 cells. Data collected over four independent repeats. Error bars are SEM. * *p* < 0.05, ** *p* < 0.01, *** *p* < 0.001.

**Figure 7 cells-11-03219-f007:**
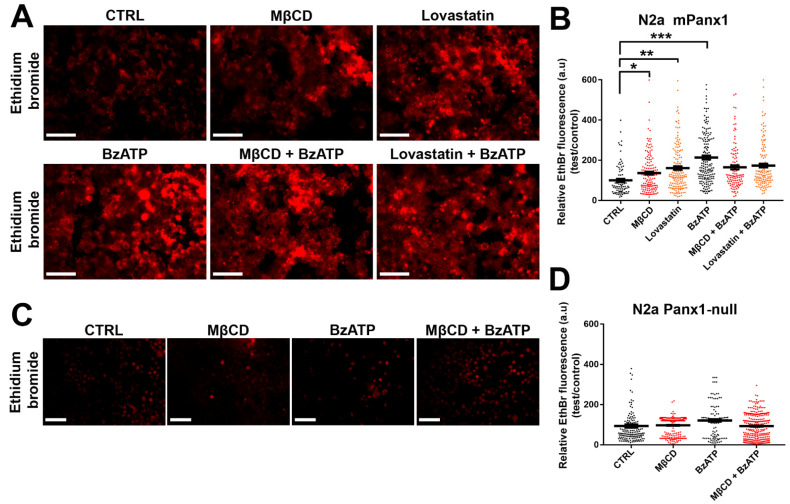
Panx1 channels mediate dye uptake in cholesterol-depleted cell plasma membrane of N2a cells. (**A**) Representative photomicrographs of mPanx1 transfected and Panx1-null. (**C**) N2a cells showing EthBr uptake after MβCD and lovastatin with and without 100 µM BzATP treatment. Scale bar: 100 µm. (**B**,**D**) Mean ± SEM values of the relative EthBr fluorescence intensity. Values were normalized to those obtained under control conditions. Each point on the graphs represents the mean values of relative EthBr fluorescence changes recorded from all cells present in a field of view obtained from 3–5 independent experiments. *p* values obtained from ANOVA followed by Dunnett’s test. * *p* < 0.05, ** *p* < 0.01, *** *p* < 0.001.

**Figure 8 cells-11-03219-f008:**
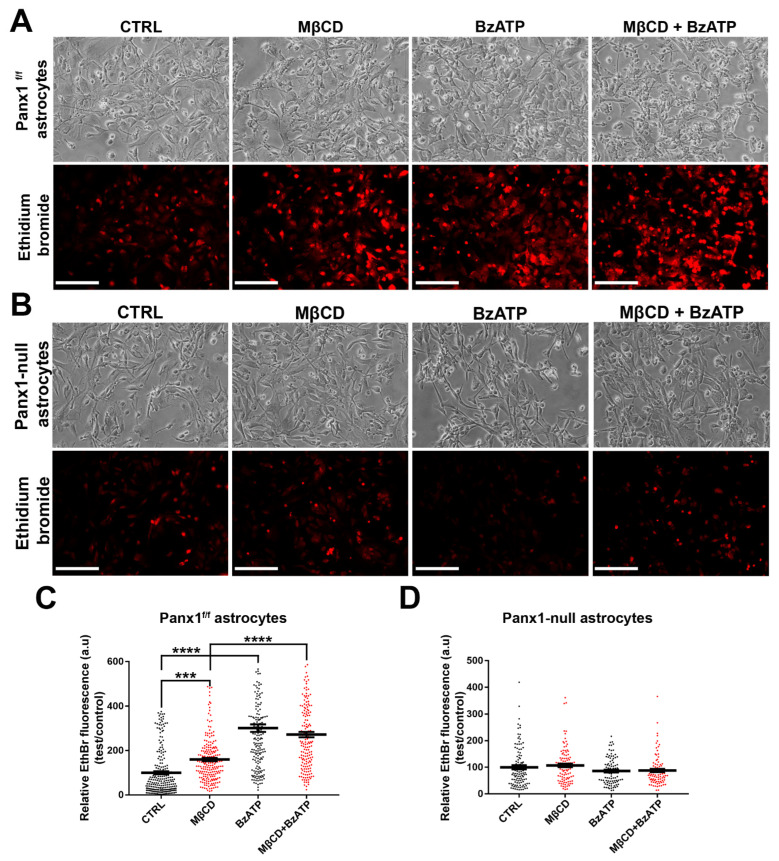
Cholesterol depletion enhances dye uptake through Panx1 channels in astrocytes. (**A**) Fluorescence micrographs of intracellular EthBr in normal conditions (CTRL) and after treatment with MβCD, BzATP, or with both drugs. Scale bar: 100 µm. (**B**) EthBr uptake in Panx1-null astrocytes induced by MβCD with and without BzATP. Scale bar: 100 µm. (**C**) Histograms showing relative EthBr uptake compared to controls for treatment with MβCD alone or together with BzATP. (**D**) Mean ± SEM values of the relative EthBr fluorescence intensity obtained from Panx1-null astrocytes after MβCD, 100 µM BzATP, and MβCD with BzATP treatment. Values were normalized to those obtained under control conditions. In (**C**,**D**), each bar on the graphs represents mean value of relative EthBr fluorescence changes recorded from all cells present in a field of view obtained from 3–5 independent experiments. *p* values obtained from ANOVA test followed by Dunnett’s test. *** *p* < 0.001, **** *p* < 0.0001.

**Figure 9 cells-11-03219-f009:**
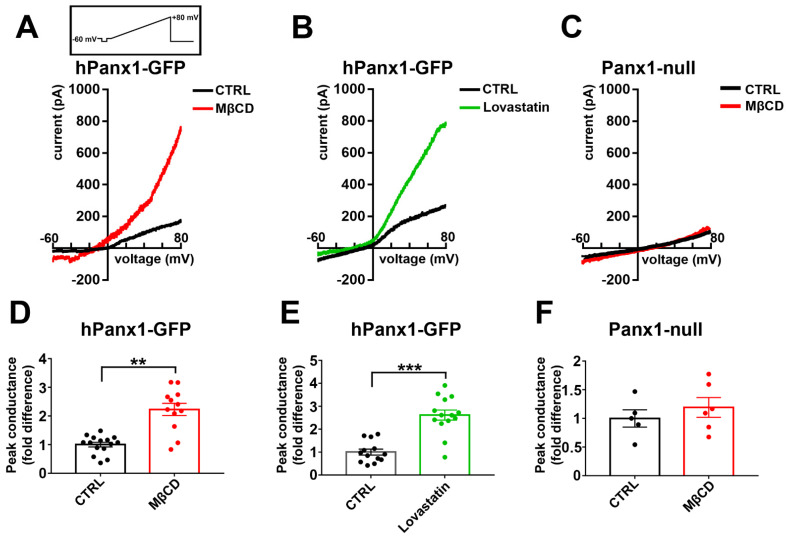
Human Panx1 conductance is activated by cholesterol depletion. (**A**,**B**) Electrophysiological recordings obtained from N2a cells expressing hPanx1 subjected to treatment with 5 mM MβCD (**A**,**D**) and 5 µM lovastatin (**B**,**E**). Note that N2a hPanx-1 transfected cells responded to cholesterol depletion with increased current amplitudes (red and green curves, in A and B, respectively) in response to 5.5 s-long voltage ramps from −60 to +80 mV compared to untreated cells, CTRL (black curves). Mean ± SEM values of the fold changes in peak conductance induced by 5 mM MβCD and 10 µM lovastatin obtained for N2a hPanx1 transfected cells are shown in parts (**C**,**D**), respectively. *p* values were obtained using Mann–Whitney tests. *n* = 10–11 cells. ** *p* < 0.01, *** *p* < 0.001. Traces in (**C**) and quantification in (**F**) correspond to recordings obtained from the Panx1null cells with and without MβCD treatment.

**Figure 10 cells-11-03219-f010:**
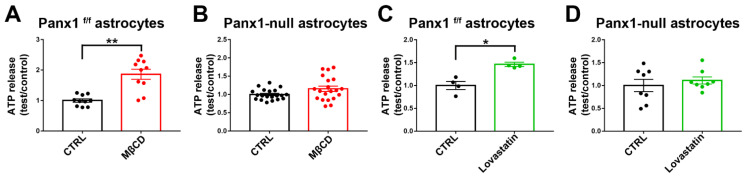
Panx1 channels mediate ATP release from cultured cholesterol depleted astrocytes. Histograms showing the ATP release from cultured Panx1^f/f^ and Panx1-null astrocytes induced by cholesterol depletion with 5 mM MβCD (**A**,**B**) and 10 µM lovastatin (**C**,**D**). Bar histogram of the mean ± SEM values of ATP release from cultured Panx1^f/f^ and Panx1 null astrocytes induced by cholesterol depletion. *p* values were obtained using Mann–Whitney tests. Data are from 2–3 independent experiments. * *p* < 0.05, ** *p* < 0.01.

## Data Availability

The data described in the manuscript are contained within the manuscript.

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
