# Peer review of "Activity and Stability of Panx1 Channels in Astrocytes and Neuroblastoma Cells Are Enhanced by Cholesterol Depletion"

_cells, 2022, doi:10.3390/cells11203219_

Round 1
Reviewer 1 Report
This is a most interesting paper with possibly important clinical significance if verified in in vivo disease models. The experiments are performed well and the data are convincing. I have a few minor comments.
pg. 8, line 327 - what concentration of MβCD was used?
pg. 9, line 352 - not clear - is this the amount of Panx1 in the medium or in the cells in the different media?
Reviewer 2 Report
This study submitted by Cibelli et al. claims that the cholesterol level regulates the distribution and dynamics of Panxs, a gap junction forming protein, on the plasma membrane. FRAP assay, dye uptake assay, and electrophysiology were used to investigate these in N2a cells and astrocytes under various temperature conditions, suggesting that cholesterol level may have influenced Panxs channel functionality, which is consequently associated with modulating neurological diseases.
It is an intriguing story that Cholesterol controls a gap junction distribution and functionality. However, However, some points need to improve in the study, which are the followings.
1. Most importantly, there is missing an important control. The authors should show and compare with negative control (e.g., a random plasma membrane protein) whether it has not same effects such as distribution and dynamics change by cholesterol. Because cholesterol has been considered an essential but general component of the membrane, it can have a general effect on the plasma membrane and its’ proteins.
2. The authors should describe more why EthBr is utilized for investigating the Panx1 channel property. Is this just for general chemicals for monitoring Panx1 channel functionality? Or a specific one?
3. Although the authors have mentioned the correlation between cholesterol and neurological disorder, I cannot find any direct relation between neural diseases and the data in this study. It would be better if the authors toned this down in the abstract.
Reviewer 3 Report
The presented manuscript aims to evaluate how cholesterol modulate the membrane properties of Panx1 channels. The authors used a combination of experimental approaches as FRAP, electrophysiological recordings, modulation of gene expression and cholesterol levels in cell cultures.
molecular dynamic simulations, Langmuir-Blodgett film, Surface Plasmon Resonance assay and various functional test to evaluate the toxicity of the peptide on cells. The manuscript is well written, the hypothesis, material & methods, results are well explained.
Major Comments:
1/ The authors should do a western-blot to quantify Panx1 expression after MβCD, lovastatin or cholesterol treatment in addition of the figure 3.
Minor Comments:
1/ I would advise to organize the abbreviations in an alphabetic order, it would be easier to read.
2/ p.1, l.36, one dot should be deleted and l.40 one is missing.
3/ p.2, l.51, a parenthesis is missing.
